# Effects of Heat Treatment and Severe Surface Plastic Deformation on Mechanical Characteristics, Fatigue, and Wear of Cu-10Al-5Fe Bronze

**DOI:** 10.3390/ma15248905

**Published:** 2022-12-13

**Authors:** Jordan Maximov, Galya Duncheva, Angel Anchev, Vladimir Dunchev, Yaroslav Argirov, Vladimir Todorov, Tatyana Mechkarova

**Affiliations:** 1Department of Material Science and Mechanics of Materials, Technical University of Gabrovo, 5300 Gabrovo, Bulgaria; 2Department of Material Sciences, Technical University of Varna, 9010 Varna, Bulgaria

**Keywords:** aluminium bronzes, Cu-10Al-5Fe bronze, heat treatment, diamond burnishing, fatigue behaviour, dry sliding wear resistance

## Abstract

Aluminium bronzes are widely used in various industries because of their unique properties, a combination of high strength, wear resistance, and corrosion resistance in aggressive environments, including seawater. In this study, the subject of comprehensive experimental research was Cu-10Al-5Fe iron-aluminium bronze (IAB) with β-transformation, received in the form of hot-rolled bars. The effects of different heat treatments (HT) and severe surface plastic deformation (SPD), conducted by diamond burnishing (DB) on the microstructure, surface integrity (SI), mechanical properties, low- and mega-cycle fatigue strength, and dry sliding wear resistance, were determined. Based on quantitative indicators, the applied heat treatments in combination with severe SPD were compared. Thus, the integral efficiency of the heat treatments was evaluated, and the heat treatments were correlated with the resulting properties and operational behaviour of Cu-10Al-5Fe IAB. For example, if the component is designed for rotational bending conditions, the combination of quenching at 920 °C in water, subsequent tempering at 300 °C for three hours, and DB provides maximum fatigue strength in both low-cycle and mega-cycle fatigue applications.

## 1. Introduction

The Cu-Al binary alloy is known as aluminium bronze. In practice, complex aluminium bronzes are used, which are denoted by the general formula Cu-Al-X, where X = Fe, Ni, Mn, Be, Co, Si, and Sn [1]. These alloys are characterised by high strength, wear resistance, and corrosion resistance in aggressive environments, including seawater. Therefore, these bronzes are the preferred construction materials for a variety of applications: gears, nuts, guides, housings and seals in valves and pumps, bushings, pistons, tubing for offshore platforms, marine engines, marine propellers, elements in the armoury industry, and bushings for sliding bearings, which are all associated with mechanical shock and cyclical loading. The effects of different elements on the specific properties and operational behaviour of aluminium bronzes are described in [1].

Cu-Al-Fe iron-aluminium bronzes (IABs) were introduced to the industry in 1870 [2] and are regarded as simpler aluminium bronzes systems [1]. IABs usually contains up to 11 wt% aluminium, where the copper forms an α-solid solution with aluminium. The increased aluminium content leads to a lower density than pure copper, which is important to consider for the practical application of this bronze. Furthermore, the presence of iron leads to grain refinement, which adds additional strength to these alloys. As the amount of aluminium increases to 4–5 wt%, the strength, hardness, and plasticity increase. With further increases in aluminium content, plasticity decreases but strength and hardness continue to increase. Moreover, IABs with Al content below 9.4 wt% exhibits a single-phase (α-phase) structure, which follows from the Cu-Al-5Fe equilibrium phase diagram [1]. These bronzes are not typically subjected to heat treatment (HT) to increase their strength and hardness. Such an increase can be achieved by cold plastic deformation. The HT of a single-phase IAB is limited to homogenisation or recrystallisation annealing. However, IABs with β-transformation (Al contents above 9.4 wt%, to which Cu-10Al-5Fe belongs) can be subjected to multiple types of HT. This two-phase bronze lacks the β-phase at the service temperature, combines high strength with good corrosion resistance, and is particularly suitable for HT. Detailed information about the effects of HT on the microstructure of complex aluminium bronzes is contained in the review paper composed by Brezina [1]. The dependence between the HT parameters and microstructure of individual types of Cu-Al-X bronze with β-transformation has been studied by numerous authors: Cu-9Al-4Fe [3,4,5], Cu-Al-Ni-Fe [6,7,8,9,10,11,12,13,14], Cu-Al-Fe-Mn [11,15,16,17], Cu-Al-Fe-Be [12]. Changing the microstructure by performing HTs on the complex aluminium bronzes also changes the mechanical properties. Table 1 shows information from the literature regarding the influence of different HT parameters on the mechanical properties of aluminium bronzes. The most frequently investigated mechanical property is hardness. The remaining properties have been investigated sporadically and information is missing for others (e.g., yield limit).

Notably, corrosion resistance is a fundamental characteristic of aluminium bronzes that determines their application in various industries. The influence of HT on the corrosion behaviour of aluminium bronzes has been studied by several authors. Aaltonen et al. [11] examined the effect of tempering on the dealloying corrosion resistance of two bronzes and established that the corrosion resistance of Cu-Al-Fe-Ni bronze is much better than that of Cu-Al-Fe-Mn bronze. Hajek et al. [15] established the optimal HT procedure for the highest possible corrosion resistance in Cu-Al-Fe-Mn bronze. Ciurdas et al. [16] studied the corrosion resistance of heat-treated Cu-Al-Fe-Mn bronze and reported that the hardened sample showed the best resistance to galvanic corrosion. Regarding other types of corrosion (selective leaching, pitting, crevice corrosion), the optimal HT was quenching at 900 °C and subsequent tempering at 550 °C. Ocejo et al. [18] investigated the effects of different types of HT on the corrosion behaviour of Cu-Al-Fe-Ni and Cu-Mn-Al-Fe-Ni bronzes in fresh water and seawater. They revealed that the major effect of HTs on corrosion was the amount and distribution of the resulting β-phase, which is prone to selective corrosion in both electrolytes.

The typical applications for aluminium bronzes are automotive, hydraulic engineering, shipbuilding, and aviation, which are all associated with slip conditions in combination with significant pressures. Therefore, sliding wear resistance is another important characteristic of aluminium bronzes, and improving the wear resistance through HT is critical for their practical application. Sadawy [14] studied the effects of HT on mass sliding wear in dry conditions of Cu-11Al-2Fe-2Ni bronze. The highest resistance was obtained after quenching at 960 °C in water, followed by ageing at 350 °C for five hours and cooling in water. The resistance was improved by the formation of fine α-phase and κ-phase grains compared to the as-cast state. Alam et al. [2] developed a novel centrifugal casting technique for the production of Cu-Al-Fe-Ni bushings. The authors found an improvement in the tribological behaviour under boundary lubrication conditions because of a reduction in the brittleness caused by the self-annealing effect. Yasar and Altunpac [19] studied the effects of ageing temperature on the mass sliding wear of two kinds of IABs under dry friction conditions. They established that quenching at 900 °C in water and ageing at 250 °C for two hours provides the maximum wear resistance. Similar studies were conducted on Cu-Al-Fe-Ni and Cu-Al-Fe-Be bronzes by Mi et al. [20]. The tribological properties (based on sliding wear tests under dry friction conditions) of these bronzes were significantly improved after heating at 950 °C for two hours, and then quenching in water at 20 °C and ageing for two hours at 450 and 350 °C, respectively. The effects of HT on the tribological characteristics of Cu-Al-Fe-Ni bronze were also studied by Muhammad et al. [21]. The specimens were heated at 930 °C for 30 min and cooled to room temperature using various methods. They showed that the furnace-cooled specimen exhibited the strongest tribological characteristics.

The fatigue behaviour is another important characteristic of aluminium bronzes in practical situations, especially when the component is cyclically loaded. In most cases, the fatigue strength of metals and alloys increases as their static strength and hardness increase. Additionally, the fatigue strength in rotary cantilever or 4-point bending tests is significantly increased by surface plastic deformation (SPD), which introduces residual compressive stresses in the surface and subsurface layers [22,23,24,25,26,27,28,29]. In principle, useful residual stresses can also be introduced into the surface layer by HT and chemical-HT, because the specific volume of the surface layer can increase, like during cold plastic deformation. Usually, the introduction of residual compressive stresses in the surface layer is accompanied by an increase in the surface micro-hardness and, thus, a synergistic effect can be observed. However, the application of SPD to improve the surface integrity (SI) of copper-based alloys has been relatively limited. Luo et al. [30] studied the micro-hardness enhancement of H62 brass by SPD with friction sliding contact using a polycrystalline diamond with a cylindrical end. Furthermore, Luo et al. [31] investigated changes in the roughness and waviness of the same H62 brass by SPD using a semi-cylindrical polycrystalline diamond. Shiou and Banh [32] studied the SPD of oxygen-free copper 101 to improve its roughness via a tungsten carbide deforming element with a spherical end. The effect of SPD on the fatigue behaviour of copper-based alloys has not been widely studied. Ezanno et al. [33] studied the fatigue strength of sand-cast ship propellers made of copper-aluminium alloy. Fatigue tests on copper alloy wire were conducted by Wang et al. [34]. The effectiveness of surface cold working, such as deep rolling, to increase the fatigue strength of quenched high-strength CuNi20Mn20 copper alloy was confirmed by rotary bending fatigue tests at a frequency of 50 Hz [35]. Liu et al. [36] demonstrated that cold rolling and subsequent annealing processes produce a recrystallised α-Cu alloy (15% Al) with ultrafine equiaxed grains (average size of 0.62 μm). This method ensures high fatigue strength of 280 MPa (107 cycles), which is much higher than the fatigue strength of 200 MPa for the nanocrystalline analogue (0.04 μm grain size), irrespective of its higher tensile strength. Gao et al. [37] investigated the effect of laser-assisted dynamic SPD on the fatigue properties of cast nickel-aluminium bronze alloy C95800. In another report, it was found that friction stir processing, which is characterised by severe plastic deformation, resulted in a more than a 40% increase in the fatigue strength of nickel-aluminium bronze [38]. The effect of SPD on the fatigue behaviour of single-phase aluminium bronze Cu-8Al-3Fe was also examined [39]. Static SPD was realised via diamond burnishing (DB). The information concerning the effects of HT on the fatigue behaviour of complex aluminium bronzes is limited [1]. Furthermore, there is a lack of reports regarding the fatigue behaviour of IABs with β-transformation as well as the effects of SPD, HT, and a combination of both on the SI characteristics, mechanical characteristics, fatigue, and wear.

This study aims to evaluate and improve the tensile strength, impact toughness, hardness, fatigue strength, and wear resistance of two-phase Cu-10Al-5Fe IAB by employing different HT techniques and static SPD.

## 2. Materials and Methods

### 2.1. Material

Commercial Cu-10Al-5Fe IAB was received as cylindrical bars and underwent chemical analysis, HTs, mechanical tests, and phase- and micro-structural analyses. Table 2 shows the chemical composition, which was determined using an optical emission spectrometer (Foundry-Master Optimum, HITACHI, Tokyo, Japan). The concentration of alloying elements was given in wt%, measured with a resolution of 0.001 wt%.

### 2.2. Specimen Treatment

For each of the tests and measurements, five groups of specimens were used, with and without SPD. Each group was subjected to the following HTs: group (1) the material was kept in as-received state (reference condition, no HT); group (2) annealing at 720 °C for three hours and furnace cooling; group (3) heating at 920 °C for one hour and quenching in water; group (4) heating at 920 °C for one hour and quenching in water, followed by tempering at 600 °C for three hours and air cooling; and group (5) heating at 920 °C for one hour and quenching in water, followed by tempering at 300 °C for three hours and air cooling. The HT parameters were chosen based on the chemical composition of the bronze, the Cu-Al-5Fe equilibrium system [1], and the recommendations given in [40].

A CCMT-120404LF KCP10 carbide cutting insert (Kennametal, Pittsburgh, PA, USA) was used for turning with the following cutting parameters: feed rate f=0.1 mm/rev, velocity $v_{c}=60\;m/\min$ and cutting depth $a_{c}=0.25\;mm$.

SPD was implemented via DB using the following process parameters: diamond insert radius r=4 mm, burnishing force $F_{b}=345\;N$, feed rate f=0.07 mm/rev, and burnishing velocity v=80 m/min. These parameters provided both low roughness and high micro-hardness which were established in our previous study [39]. DB was carried out after the HTs. The schematic of the DB set-up and the methodology were explained in our previous studies [39,41,42].

### 2.3. Microstrucutre

The phase structure analysis was performed by X-ray diffraction (XRD, Bruker D8 Advance diffractometer, Karlsruhe, Germany). Crystallography Open Database was used to determine the peak positions. The microstructures of the cylindrical bars’ cross-sections were observed by scanning electron microscopy (SEM, LYRA I XMU, Tescan, Brno, Czech Republic), after polishing and etching the specimens using 20% FeCl_3_ solution.

### 2.4. Mechanical Characteristic

#### 2.4.1. Tensile Test

The tensile strength, yield stress, and elongation were determined from tensile tests at room temperature by means of a Zwick/Roell Vibrophore 100 testing machine (Ulm, Germany) with a strain rate of $10^{-3}\;s^{-1}$. The sizes of the specimens are depicted in Figure 1a, used according to [43]. Five groups of specimens according to Section 2.2 were evaluated without DB. Each group contained three specimens. The result for each group was established using the arithmetic mean obtained from three specimens.

#### 2.4.2. Impact Toughness

The impact toughness was evaluated according to [44] using a Charpy Universal Impact Tester with a maximum impact energy of 300 J. The sizes of the specimens are depicted in Figure 1b. Five sample groups, each consisting of three samples, were evaluated without DB. The result for each group was established using the arithmetic mean obtained from three specimens.

#### 2.4.3. Hardness

The hardness was measured according to [45] using a ZWICK/Indentec—ZHVμ-S hardness tester (Ulm, Germany) with spherical-ended indenter (D = 2.5 mm), loading F = 63 kg, and holding time 10 s. The sizes of the specimens are depicted in Figure 1c. One specimen without DB from each of the five groups was assessed using five measurements. The value for each group was established as the arithmetic mean of the five measurements.

### 2.5. Surface Integrity

The roughness parameter $R_{a}$ was measured by a Mitutoyo Surftest SJ-210 surface roughness tester (Takatsu-ku, Kawasaki, Japan). The final roughness value $R_{a}$ for each sample was calculated as the arithmetic mean of five measurements of five generatrixes at a 72° angle.

The HV0.05 surface micro-hardness measurements were performed as previously described [46]. We used a ZHVμZwick/Roell micro-hardness tester (Ulm, Germany) with automated processing of the measurement results by a 0.05 kg load and a 10 s holding time. The sizes of the specimens are depicted in Figure 1c. Two samples were measured from each of the five sample groups. The second sample from each group was subjected to DB. Twenty measurements were conducted for each specimen. The final value of the surface micro-hardness corresponds to the grouping centre.

A vertical θ/θ X’Pert PRO MPD diffractometer (Billerica, MA, USA) with a pin-hole collimator measuring 1×0.5 mm in the primary beam was used to measure the residual stress profile. The X-ray tube’s mode of operation (high voltage/current) was 35kV/22mA. The $\sin^{2}\psi$ method with a least-squares fitting procedure was used to evaluate the residual stresses. The measured diffraction profile of α-Cu {311} plane has its maximum at 2θ≈144.5° for the filtered CrKα radiation used. Diffraction profiles were determined using the centre of gravity method, and the lattice deformations were calculated. For the generalised Hooke’s law, the Winholtz and Cohen method and the X-ray elastic constants $s_{1}=-3.31\;TPa^{-1}$ and $\frac{1}{2}s_{2}=12.45TPa^{-1}$ were applied. The X-ray experiment used the following parameters: 2θ range of 134−154°, 2θ step of 0.2° and tilt defined by $\sin^{2}\psi$ = 0, 0.15, 0.3, 0.45, and 0.6 for both positive and negative values of angle ψ. The effective penetration depth of the MnKα radiation was approximately 2.5−4.5 μm.

Based on the chemical composition and phase analysis (over 80% α-Cu), the residual stress distribution could only be measured on the α-Cu phase. To analyse the stress distribution in a depth, the surface layers were gradually removed by electrolytic polishing using a PROTO Electrolytic Polisher (Proto manufacturing Inc., Taylor, MI, USA) with electrolyte type E5.

### 2.6. Operating Behaviour

#### 2.6.1. Fatigue Tests

Two sets, one without DB and one with, of hourglass-shaped fatigue specimens were manufactured on an Okuma lathe. The geometry (given the requirements of the used UBM fatigue testing machine) of the hourglass-shaped samples is depicted in Figure 1d. Each of the sets contained five groups of samples according to Section 2.2. The second set was subjected to DB, using DB parameters described in Section 2.2. A lubricant-cooled Hacut 795-H was used for both turning and DB.

Rotating cantilever bending fatigue tests were carried out (Figure 2). The cycle asymmetry factor was R = −1. The loading frequency was 50 Hz in air. The accuracy for counting the number of cycles to failure was within 100 cycles. The rotating load magnitude was controlled by means of a lever system. The stress amplitude was calculated as follows:(1)$$\sigma =\frac{32M_{b}}{\pi d_{\min}^{3}}$$
where $M_{b}=PL$ is the bending moment, P is the assigned force, L is the length of the cantilever part, $\pi d_{\min}^{3}$ is the bending resistance moment of the specimen cross-section, and $d_{\min}$ is the minimum diameter of the fatigue sample. The accuracy of setting the force P was ±1 N and of the length L was ±0.1 mm. The maximum error Δσ of the stress amplitude was 2.44 MPa. For each stress amplitude, one specimen was tested until failure or unacceptably large plastic deformation was reached, leading to the automatic shutdown of the testing machine.

#### 2.6.2. Sliding Wear Tests

The effect of DB on the sliding wear resistance of IABs in the as-received state for two friction modes (boundary lubrication and dry friction) was studied in [42,47]. DB significantly improves the wear resistance of IABs. Therefore, the present study focused on the sliding wear behaviour of a Cu-10Al-5Fe bronze-hardened steel tribo-system in a dry friction condition and, more specifically, the effects of the HTs on the IAB’s sliding wear resistance. The experimental study of the bronze-steel tribo-system was conducted according to the fixed steel segment-rotating bronze roller kinematic scheme. Figure 3 shows the sliding wear test experimental setup.

Medium carbon C45 steel was chosen as the material for the counter-body and was subjected to a three-stage HT to achieve a tempered martensite structure which ensures high hardness (at the expense of reduced impact toughness) [47]: (1) normalising (austenitising at 880 °C for two hours and subsequent air cooling) to achieve grain refinement; (2) hardening (austenitising at 840 °C and quenching in water) to achieve a martensitic structure (fine-needle martensite, also called structureless martensite); (3) tempering at 180 °C to achieve tempered martensite structure with a hardness of HRC 56. Thus, the requirements for the combination of relatively high hardness and strength were satisfied. After the final grinding, the inner surface roughness was $R_{a}=0.32\;\mu m$.

The repeatability of the experimental data for mass wear was evaluated via selecting three specimens from each sample group. The cylindrical surface (with radius R = 11.5 ± 0.05 mm and width c = 16 ± 0.15 mm) of the bronze sample is in contact with counter body’s inner cylindrical surface. The bronze specimen rotates at a constant frequency of n = 1350 min^−1^. These parameters provide sliding velocity v = 1.63 m/s. The normal load P = 95 N (Figure 3) was applied to the contact spot centre of gravity between the bronze sample and the steel counter body and was set via a lever system in the load beam. The nominal contact area A_a_ for each specimen is A_a_ = R φ c ≈1.6057 × 10^−4^ m^2^, and the nominal contact pressure is p_a_ = P/A_a_ = 0.591 MPa. The specified nominal contact pressure was conformed to the dry friction condition.

Considering the significant amount of heat generated, the mass wear of the bronze samples under dry friction conditions was measured consequently after intervals of 5, 10, and 15 min for the sliding friction path lengths of 67.74, 101.61, and 135.48 m, respectively. The mass wear of the samples was measured for a given friction path at constant load and sliding speed. The methodology includes the following succession: the initial mass m_0_ of the bronze specimen was measured to the nearest 0.1 mg via a WPS 180/C/2 electronic balance and before each measurement, the bronze specimen was cleaned to remove mechanical and organic particles and dried with ethyl alcohol to prevent electrostatic effects.

## 3. Results and Discussion

### 3.1. Effect of the HT and DB on the Microstructure

Figure 4 shows a section of the Cu-Al-Fe equilibrium phase diagram with a 5% Fe plane [1]. Below the eutectoid line, the Cu-rich α-phase, the electronic compound $\gamma^{\prime }\to Cu_{9}Al_{4}$, and the intermetallic compound $\delta_{Fe}\to Fe_{3}Al$ are formed. The content of Fe in the investigated bronze is 5.7% (see Table 2). Therefore, the relative share of finely dispersed $Fe_{3}Al$ particles is greater. They prevent the growth of α-grains at high temperatures, increase the α-phase strength, eliminate the phenomenon of spontaneous annealing that increases brittleness, and block the growth of the $\gamma_{2}-$ phase that precipitates in the form of large plates and cause brittleness.

The phase analysis results are shown in Figure 5. The α-phase peaks were shifted to smaller angles compared to those calculated for pure Cu due to the dissolved aluminium atoms that deform the α-copper crystal lattice.

For the group 1 and 2 samples, the eutectoid line (see Figure 4) is crossed upon cooling at a subcritical speed, i.e., the phase recrystallisation mechanism is diffusion, and the electronic compound $\gamma^{\prime }\to Cu_{9}Al_{4}$ was registered, since eutectoid decomposition $\beta \to \alpha +\gamma^{\prime }$ took place. Due to its high dispersion, the intermetallic compound $Fe_{3}Al$ was not observed. For the remaining groups (groups 3 to 5), recrystallization is diffusion-free upon cooling at 920 °C in the water phase, and a martensite-like $\beta^{\prime }-$ phase ($Cu_{3}Al$) was registered, which is a product of deformation transformation of the supereutectoid β− phase ($Cu_{3}Al$). The highly dispersed compound $Fe_{3}Al$ is not registered. Notably, a fully martensitic structure cannot be obtained after quenching at 920 °C in water due to the high content (5.7 wt%) of Fe and the negligible amount of the β-stabilising element Ni. The α-stabilizing Fe suppresses the martensitic transformation and favours the formation of bainitic structure [1].

The microstructure of Cu-10Al-5Fe is shown in Figure 6. For groups 1 and 2 (Figure 6a,b), the structures exhibit a diffusion mechanism of phase recrystallisation, consisting of equiaxed (30–40 μm) crystals of α—Cu solid solution. Eutectoid colonies, a product of the breakdown of the β-phase ($\beta \to \alpha +\gamma^{\prime }$), as well as the intermetallic compound $Fe_{3}Al$ in the form of highly dispersed particles in the α-Cu, are observed at the grain boundaries. Therefore, the as-received IAB bar is likely composed of three phases. For the samples annealed at 720 °C (Figure 6b), grain enlargement is observed with both 60–80 μm and sub 20 μm grain sizes. It can be concluded that secondary recrystallisation took place. The eutectoid colonies have dissolved in the α-grains and are observed as $\gamma^{\prime }$-precipitates at the boundaries of the large α-grains or within the grains themselves.

As a consequence of the mechanical impact (turning and DB), two zones are distinguished: zone 1 (direct contact) and zone 2 (zone of deformation influence). The latter is observed only for DB specimens. The zones are marked with a dashed yellow line. For the as-received group (Figure 6a), zone 1 has a depth of 25 μm, and the sliding lines for only turned specimens are rougher, as a result of metal detachment from the surface. Zone 2 is outlined for DB specimens, which is visible at high magnification and is characterised by the presence of twinning grains. For the annealed turned specimens (Figure 6b), the sliding lines are relatively rough, a consequence of cutting relatively soft metal. For the DB specimens, zone 1 has finer sliding lines and a sliding strip is observed. Due to the significantly greater plasticity of the annealed specimens, there is a possibility of significant strain hardening of the grains. The propagation of the deformation wave significantly increases the depth of zone 2, in which sliding lines and twinning grains are observed.

The microstructure of group 3 samples quenched at 920 °C in water is bainite-like (Figure 6c), with elongated plate grains. The main phases after phase recrystallisation are the Cu-rich α-phase and $\beta^{\prime }-$ phase (see Figure 5). After turning, zone 1 is observed, containing partially affected grains from sliding lines that are not completed at the boundaries of the entire grain. No sliding occurred in high-hardness grains. The observed bainite-like structure is not prone to cold-hardening, but partially reveals its effects. The microstructure of DB specimens (Figure 6(c3)) shows the absence of zone 2. In zone 1, sliding lines exist in a small local area. A deformation of the surface grains is observed in the direction of the relative movement of the deforming diamond insert toward the treated surface.

Figure 6d shows the microstructure of the samples from group 4 (quenched at 920 °C in water and tempered at 600 °C). The low temperature (600 °C) of repeated phase recrystallisation, which is very close to the eutectoid line (565 °C)—see Figure 4, is the reason for the formation of two structural groups. One is bainite-like $\left(\alpha +\beta^{\prime }\right)$-phases, and the other is composed of distinct α-Cu grains that have not undergone phase recrystallisation. After turning, zone 1 is observed with a depth of approximately 25 μm. DB forms two zones: a narrow zone 1 with highly deformed grains in the direction of the relative movement of the diamond insert, and zone 2 with characteristic sliding lines, a consequence of a deformation wave from the surface layer.

Figure 6e shows the microstructure of the specimens from group 5 (quenched at 920 °C in water and tempered at 300 °C). The phase composition is a bainite-like $\left(\alpha +\beta^{\prime }\right)$ mixture with a distinct $\beta^{\prime }-$ phase. A greater dispersion of grains is observed compared to the specimens from group 3 (only quenched at 920 °C in water). The mechanical impact from turning and DB causes only deformation of the surface layer, without the manifestation of the sliding mechanism, and this effect is significantly more pronounced after DB (Figure 6(e3)).

### 3.2. Effect of HT on the Mechanical Characteristics

Figure 7 shows the effects of HT on the mechanical characteristics obtained via tensile tests. It should be noted that the as-received state is not equivalent to the as-cast state. From Figure 7 in the as-received state, the material exhibits high tensile strength and high yield strength but has the lowest elongation and smallest plasticity. Therefore, the material experienced significant hot-mechanical strengthening in the process of processing the as-received bar. It is known that consecutive hot (at 850 °C) forging and rolling of nickel-aluminium bronze significantly increases its yield limit and tensile strength, but greatly reduces the elongation [48]. A similar result was obtained by Ma et al. [49] for hot-forged Cu-10Al-5Fe-5Ni-0.2Mn commercial nickel-aluminium bronze, subjected to heavy hot rolling without subsequent annealing. In other words, in the production process of Cu-10Al-5Fe bars, the material has undergone hot-mechanical strengthening due to transformation strengthening. The latter is caused by an increase in the volume fraction of the β-phase transformation product [50]. All HTs caused recrystallisation of the material, increasing the plasticity, quantitatively expressed by the elongation. The greatest ductility was obtained after annealing. This is due to the finer and rounded grains forming a more homogeneous microstructure (see Figure 6).

In general, with an increase in plasticity, a decrease in the yield limit is observed. All HTs, except annealing, increased the tensile strength, with the highest static strength obtained after quenching. The resulting structure is bainite-like (see Figure 6), characterised by high static strength. This structure of the aluminium bronzes is sensitive to internal notches [1], which is essential under dynamic loading. Under static loading, these internal concentrators are not relevant, as a redistribution of internal stresses occurs. The subsequent tempering reduces the tensile strength at the expense of increased plasticity. This tendency is more pronounced at a lower ageing temperature, i.e., the elongation decreases with increasing temperature, the yield limit increases, and the tensile strength decreases. This trend of the elongation and the tensile strength was also found by Mi et al. [12] for nickel and beryllium aluminium bronzes.

Figure 8 shows the effects of HT on the hardness obtained. Annealing maximises ductility (see Figure 7) but minimises hardness. The hardening (heating at 920 °C and quenching in water) of the as-received state leads to a lower hardness than the original. Quenching at 920 °C is preceded by recrystallisation of the bronze, which eliminates the effect of mechanical hardening obtained in the process of making the workpiece, increases the ductility (see Figure 7), and reduces the hardness (see Figure 8). The subsequent tempering increases the hardness depending on the temperature, varying with the decomposition of the saturated solid solution (β-phase). Tempering at 300 °C provides maximum hardness. The probable reason is that the decomposition of the solid solution is completed during the second stage, known as dispersion hardening or strengthening tempering [51]. The decreasing hardness with increasing ageing temperature of the IAB confirms the results obtained by Aaltonen et al. [11] and Mi et al. [12] for nickel and beryllium aluminium bronzes. This trend is also confirmed by Sadawy [14].

Figure 9 shows the effects of HT on the impact toughness, as well as corresponding fracture surfaces. The material has the smallest impact toughness in the as-received state due to the mechanical strengthening in the process of making the workpiece, resulting in increased brittleness: high strength and hardness, but low plasticity and toughness. All HTs increased the impact toughness. The largest increase (almost three-fold) was obtained after quenching and subsequent high-temperature ageing (tempering), and the smallest increase was observed after quenching and subsequent low-temperature ageing. The main reason is that low-temperature ageing maximises the hardness (see Figure 8), but the brittleness increases due to the hard β’-phase presence (see Figure 6e). The increasing impact toughness with increasing ageing temperature confirms the results obtained by Aaltonen et al. [11] for two types of aluminium bronze—nickel and beryllium.

Although the fracture is usually brittle under impact loading, the fracture surfaces (Figure 9) illustrate some specifics of the impact fracture mechanism depending on the HT. For example, after annealing, the microstructure is characterised by marked homogeneity. As a result, the fracture surface is planar and the fracture is typically brittle. After quenching, the microstructure (coarse-grained bainite-like) exhibits pronounced inhomogeneity (see Figure 6c). Because of this, the fracture surface has a highly pronounced relief, and the mechanism of destruction is mixed—there are areas with brittle destruction, in addition to ductile-plastic destruction.

### 3.3. Effects of HT and DB on the SI

The initial roughness (before DB), estimated with the $R_{a}$ parameter, is in the range of 0.5–0.7 μm. After DB, the roughness is greatly reduced to the range of 0.10–0.15 μm.

The influence of the HT and DB on the micro-hardness $HV_{0.05}$ is shown in Figure 10. The large scattering of all samples (as-received and heat treated) before DB is observed. Severe SPD via DB homogenises the structure of the surface layer and nearby subsurface layers and increases their specific density (see Figure 6), which significantly reduces the scattering. Except for the samples of groups 3 and 5, all other samples increased their surface micro-hardness after DB. The greatest increase was obtained when the specimens were in as-received state. Because of the significantly larger radius of the diamond insert compared to the feed rate, the DB process causes cyclic loading of the surface layer points due to the inevitable overlapping. Soft or annealed metals tend to harden towards a stable limit, and initially hardened metals tend to soften [52]. For this reason, the samples from groups 3 and 5, which contain the hard β- and γ-phases, showed a softening effect of the surface layer after DB, i.e., they have reduced surface micro-hardness.

Figure 11 shows the axial and hoop residual stress distribution after turning and after DB, respectively. As confirmed previously, the distribution of the axial residual stresses is characterised by a more intense compressive field near the treated surfaces than that found in the distribution of the hoop residual stresses [47]. Notably, the residual stress fields of the heat-treated non-DB sample groups are a superposition of three states: (1) as-received, (2) after HT, and (3) after turning. The residual stress field of the reference condition is a superposition of two states: (1) as-received and (2) after turning. The received bars are produced via hot plastic deformation which can be considered as a sequence of cold plastic deformation (introducing strain hardening, respectively residual macro-stresses) and recrystallisation annealing (eliminating strain hardening, respectively residual macro-stresses). The HT groups were subjected to temperatures exceeding the recrystallisation temperature of Cu-10Al-5Fe bronze. Therefore, the residual stresses in the surface and nearby subsurface layers of all non-DB specimens are introduced by turning. The turning with the corresponding geometry of the used cutting wedge introduces axial and hoop residual compressive stresses at depths of 0.2−0.3 mm from the surface layer. Since the turning was performed with the same cutting tool and with the same cutting modes, the different distribution of the residual stresses for the individual groups of samples is due to the different microstructure obtained by the respective HTs. DB plastically deforms the surface and nearby subsurface layers, drastically reducing surface roughness. As a result of the introduced plastic deformation, a redistribution of the residual stresses occurs after turning. The residual stresses increase significantly in absolute value and the depth of the compressive zone increases significantly—over 0.7 mm.

### 3.4. Effects of HT and DB on the Fatigue Behaviour

Figure 12 shows the S-N curves in a double-logarithmic coordinate system. All HT non-DB samples show lower fatigue strength compared to group 1 samples due to reduction of the strengthening effect in as-received bars. The lowest fatigue strength is observed in samples subjected to annealing (group 2), revealing the lowest tensile strength and hardness (see Figure 7 and Figure 8). After quenching and tempering (groups 4 and 5), the fatigue strength increased compared to group 3, which was only subjected to quenching. The reason is the fine-grained bainite-like structure (see Figure 6d,e). Group 5 combines maximum hardness and surface micro-hardness with high plasticity and tensile strength. In the low-cycle region, group 5 shows greater fatigue strength, and the slope of the curves shows that, under $10^{4}$ cycles, group 5 outperforms the reference condition (group 1). Conversely, group 4 has the smallest slope and, in the mega-cycle fatigue region, shows greater strength compared to groups 3 and 5.

DB improves the fatigue strength of all sample groups, a consequence of the compressive residual stresses (see Figure 11). After DB, group 5 showed greater fatigue strength compared to DB group 1. The improvement is significant for low-cycle fatigue. Conversely, after $10^{6}$ cycles, there was a tendency for group 1 to show higher fatigue strength compared to group 5.

Figure 13 shows the dependence of fatigue strength on yield limit as a function of fatigue life. Group 3 has the highest yield limit, but the fatigue strength in both low-cycle and high-cycle fatigue is smaller compared to those of groups 1, 4 and 5. This phenomenon can be explained by the increased internal notch sensitivity [1], which is observed in group 3.

Figure 14 illustrates fatigue strength improvement by DB for low-cycle and mega-cycle fatigue, according to the number of cycles to failure for all five groups. Except for group 4, the HT increases the benefit of DB for increasing fatigue strength. Group 5 shows the greatest improvement. The main reason is the introduced useful residual hoop and axial stresses, which have the largest absolute value and maximum depth for group 5.

The fractographic analysis was conducted based on the fatigue surface fracture of one specimen from each of the five groups (without DB), subjected to cyclic bending with an amplitude of the bending stress in the interval 480–520 MPa. These values exceed the yield strength of the corresponding bulk material subjected to a corresponding HT. Since the samples are not subjected to DB, the maximum tensile stress occurs in their surface layer, where the fatigue micro-cracks nucleate (Figure 15). Due to the cyclic loading with controlled deformation in the plastic region (for the surface and subsurface layers), the evolution of intrusion–extrusion leads to the formation of fatigue micro-cracks.

Figure 15a shows the dynamic zone of the surface fracture for the as-received specimen. Two areas of the cyclic fatigue zones, A and B, formed as a result of the intrusion-extrusion mechanism. The directions of the cyclic deformation waves in the two zones diverge, and as a consequence, a narrow intermediate zone is formed between them. The intermediate zone favours the formation and development of a fatigue macro-crack. The local surface region of fatigue micro-crack nucleation (Figure 15(a2)) includes an intermediate zone, a surface zone with local development of intrusion, a zone with cellular (transcrystalline) destruction and an intercrystalline destruction zone.

The dynamic zone of the fracture surface of the annealed sample is characterised by a weak relief (Figure 15(b1)) as a consequence of the homogenised microstructure by repeated recrystallisation. Several centres of fatigue micro-crack nucleation are observed. Figure 15(b2) shows the characteristic extrusion deformation mode for the surface and subsurface layers in the dynamic zone. The wave deformation process takes place with larger amplitude, a consequence of the greater plasticity and the absence of deformation barriers, such as twinning grains and separated eutectoid regions. Near the surface, first-mode micro-cracks are noticeable, as well as the Fe_3_Cu intermetallic compound.

In the surface fracture of the specimen quenched at 920 °C in water, the dynamic and static zones are delineated (Figure 15(c1)). From higher magnification of the dynamic zone around the surface layer (Figure 15(c2)), the failure mechanism appears to be mixed, due to the bainite-like structure with coarse plate-like grains (see Figure 6c). Local plastic areas (extrusion) are observed, as well as local areas with brittle destruction, in which a significant number of micro-cracks and those at grain boundaries are formed.

Figure 15d shows the dynamic surface fracture zone of a specimen from group 4. This group of specimens is characterised by a bainite-like microstructure with smaller rounded grains (see Figure 6d). In the subsurface layers of the dynamic zone, numerous micro-cracks are observed, as well as secondary cracks formed between two areas of extrusion. The multitude of micro-cracks hinders the growth of the macro-crack, due to the internal stress distribution.

Thus, the group 4 specimens show a higher mega-cycle fatigue strength. The observed pattern was obtained at a bending stress amplitude of 500 MPa. As the amplitude increases, the density of micro-cracks increases, and hence multiple macro-cracks are formed. In other words, the fatigue strength in the low-cycle field will decrease which is confirmed by the smaller slope of the fatigue curve of the samples from group 4 (see Figure 12).

Figure 15e shows part of the dynamic surface fracture zone of a group 5 specimen, quenched at 920 °C in water and tempered at 300 °C. The microstructure of the samples from this group is bainitic-like with plate grains, but significantly smaller than the grains of the samples from the group 3 (quenched only). This microstructure predetermines a mixed destruction mechanism in the dynamic zone immediately adjacent to the surface layer (Figure 15(e1)). The surface zone A is characterised by pronounced extrusion and grain boundary destruction and secondary cracks are observed in the same area. This region borders an intermediate region of quasi-brittle destruction.

### 3.5. Effects of HT on the Sliding Wear Resistance

The mass wear of the five groups of non-DB specimens is shown in Figure 16. The low scattering is a criterion for reliable repeatability of mass wear results. For the investigated time interval, the hardened and tempered specimens (group 5) show the greatest mass wear, and the annealed specimens (group 2) have the least wear. Compared to the annealed specimens, the percentage increase in mass wear at the final friction path is as follows: group 1—16.8%, group 3—30.7%, group 4—4%, group 5—55.4%. The mass wear increases with increasing hardness of the sample groups (see Figure 8). The results agree with the study in [21], where the least mass wear of Cu-10Al-5Ni-4Fe bronze was observed in the samples subjected to annealing.

For Cu-10Al-5Fe bronze, the maximum hardness was obtained for group 5 samples, but the maximum mass wear was obtained for the same HT. These results agree with the conclusions made in [19] for IABs, and the probable cause of this phenomenon is the presence of a hard intermetallic phase, which is obtained after tempering (ageing) at 300 °C. These hard precipitate particles can tear the surface of the Cu-matrix during the sliding wear test [19].

The contact temperature change for the first samples from each group is shown in Figure 17 for the three time intervals: 0–5, 5–10, and 10–15 min. The sample subjected to quenching and subsequent tempering at 300 °C shows the lowest temperature in the three time intervals. Therefore, this HT provides the lowest sliding friction coefficient (based on Coulomb’s theory) in the bronze-hardened steel pair. This result is compatible with [14] for Cu-Al-Fe-Ni bronze quenched and ageing at 350 °C, where the counter body is 600 grit SiC paper, fixed on an aluminium wheel. The measured average temperature in the bronze roller-steel segment contact increases with increasing ageing temperature.

This confirms the conclusion made by Sadawy [14] about the wear behaviour of heat-treated nickel-aluminium bronze.

The tribological behaviour of a friction pair is determined by three factors: lubrication, friction, and wear. In the case of intense dry sliding friction, the first factor is not considered. The bulk hardness by itself is not a sufficient criterion to explain the tribological behaviour of the studied bronze. For the Cu-10Al-5Fe bronze–hardened steel friction couple, increasing the hardness of the bronze reduces the friction coefficient, but increases mass wear. The combination of the specific concentration of individual phases, grain size and orientation, bulk hardness, and micro-hardness can provide a more insight into the tribological behaviour.

The graphs in Figure 17 provide information not only for the generated heat due to friction, but also the wear mechanism. The smooth curves and relatively low temperatures for the samples from groups 3 and 5 indicate a uniform and continuous wear process. In contrast, the curves for groups 1, 2, and 4 show a pronounced non-uniform friction, demonstrated by a significantly higher temperature. The uniform wear and the relatively small amount of heat generated suggest a predominantly abrasive wear mechanism. The presence of fluctuations in the temperature curves and the significant heat generated due to friction indicate a mixed wear mechanism—abrasive wear and adhesive wear mechanisms. The morphology of the worn surfaces shown in Figure 18 reflect the behaviours observed in the temperature curves. On the worn surfaces of the samples from groups 1, 2, and 4 (Figure 18a–d), distinct oxidised zones are observed due to the significant heat generated at high temperatures (see Figure 17). The high contact temperature in the dry sliding condition, favoured by the lower hardness, leads to local adhesive pits. Abrasive scratches are observed in the remaining regions. Therefore, for samples from the groups 1, 2, and 4, the wear mechanism is a mixed adhesive and abrasive wear mechanism. Figure 18c,e shows that the abrasive wear mechanism is dominant the specimens of the groups 3 and 5.

## 4. Integral Efficiency of Heat Treatment Types of Cu-10Al-5Fe Bronze

A comparison of the different HTs in combination with severe SPD is shown in Table 3. It is assumed that the comparison criteria have the same relative weights. The ranking (from 1 to 5 where 1 is the highest rank and 5 is the lowest) is based on the quantitative results. The effect of severe SPD is considered in the seventh criterion—fatigue strength. All HT groups outperform the group 1 (as-received reference condition). The absence of first-position ranking for as-received specimens indicates the necessity and usefulness of proper HT. Depending on the application, an appropriate HT can be selected from Table 3 (bold numbers). For example, if the component operates under rotational bending conditions, the combination of quenching at 920 °C in water, subsequent tempering at 300 °C for three hours, and DB provides maximum fatigue strength in both low-cycle and mega-cycle fatigue applications. The maximum impact toughness is provided by a similar HT, but the tempering (ageing) is conducted at 600 °C.

The integral efficiency of the different HTs is shown in the last row of Table 3, and the group 5 samples have the highest ranking. It can be concluded that the key to optimal HT according to a predetermined criterion is the temperature and holding time of the tempering after quenching.

## 5. Conclusions

The aim of this article is to quantify the effects of different HTs and severe SPD on the microstructure, SI, tensile strength, impact toughness, hardness, fatigue strength, and dry sliding wear resistance of Cu-10Al-5Fe IAB. An experimental approach was used, including XRD and SEM analyses, an evaluation of the SI obtained after different HTs and DB, tensile tests, impact toughness and hardness tests, rotating cantilever bending fatigue tests, and dry sliding wear resistance tests. Based on the experimental results, a ranking from 1 to 5 was determined for the four different HTs and the as-received condition according to nine criteria with equal relative weights. The integral efficiency of the five groups was obtained by summing the ranks according to the individual criteria. The major new findings can be expressed as follows:All HT groups outperform the first group (as-received). The absence of first-position ranking for as-received group confirms the necessity and usefulness of the proper HT. When the bronze component operates under rotational bending conditions, the combination of quenching at 920 °C in water, subsequent tempering at 300 °C for three hours, and DB provides maximum fatigue strength for both low-cycle and mega-cycle fatigue. Opposite trends were found for friction and mass wear in the heat-treated bronze-hardened steel tribo-system under dry sliding wear behaviour. (1) When the requirement is for minimal friction to minimise the generated heat, quenching at 920 °C in water and subsequent tempering at 300 °C for three hours is most suitable. (2) When the goal is to minimise the mass wear of the bronze specimen, the most suitable HT is annealing at 720 °C.Maximum static strength is provided after quenching at 920 °C in water, and maximum plasticity after annealing at 720 °C. For bronze components operating under impact load conditions, the most suitable HT is quenching at 920 °C in water and subsequent tempering at 600 °C for three hours.Quenching at 920 °C in water and subsequent tempering at 300 °C for three hours provides the maximum integral efficiency for the HT of Cu-10Al-5Fe IAB and therefore it is appropriate for wider industrial applications.

## Figures and Tables

**Figure 1 materials-15-08905-f001:**
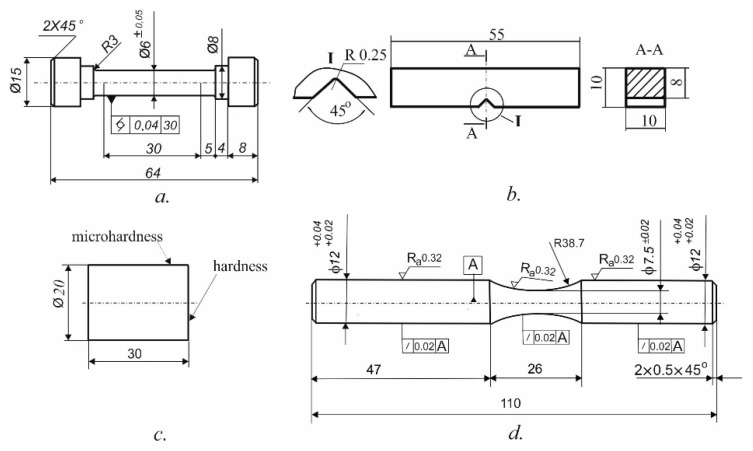
Geometry of the samples: (**a**) tensile test; (**b**) rotating bending fatigue test; (**c**) Charpy (impact toughness) test; (**d**) rotating cantilever bending fatigue test (the sizes are in mm).

**Figure 2 materials-15-08905-f002:**
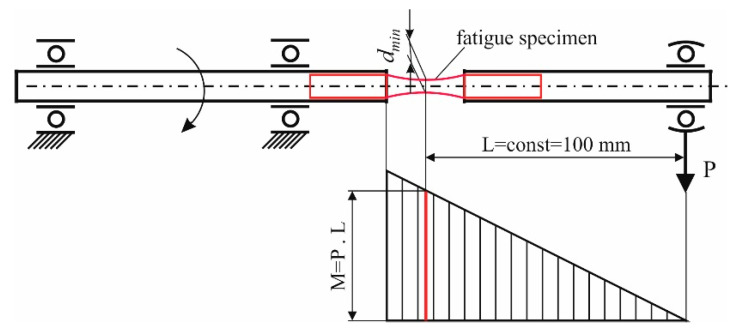
Rotating cantilever bending fatigue test scheme.

**Figure 3 materials-15-08905-f003:**
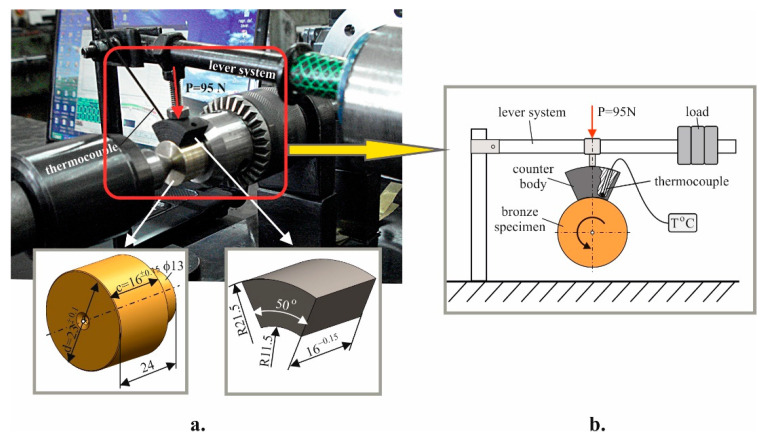
Photo and geometry of the specimen and counter-body (**a**), and scheme (**b**) of the tribo-tester.

**Figure 4 materials-15-08905-f004:**
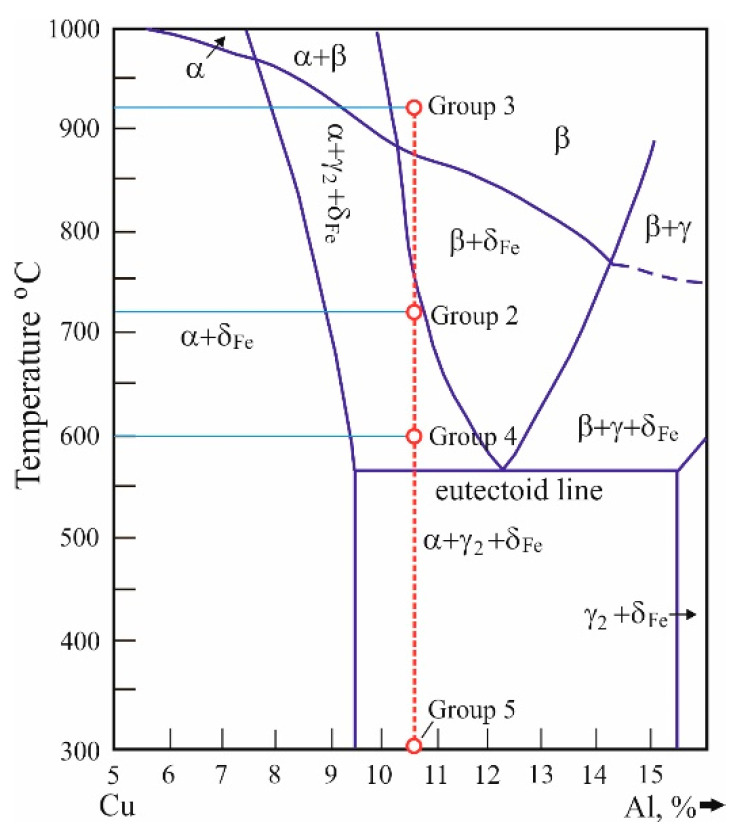
A section of the Cu-Al-Fe equilibrium phase diagram with a 5% Fe plane [1].

**Figure 5 materials-15-08905-f005:**
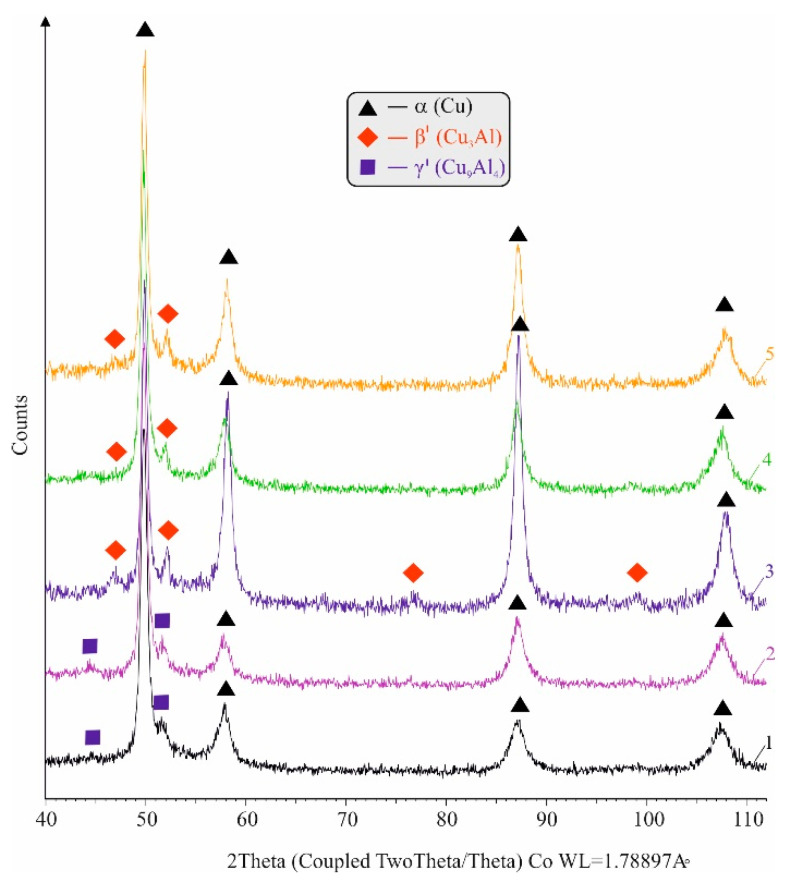
Phase analysis results for Cu-10Al-5Fe bronze: 1—as received; 2—annealed; 3—quenched; 4—quenched and tempered at 600 °C; 5—quenched and tempered at 300 °C.

**Figure 6 materials-15-08905-f006:**
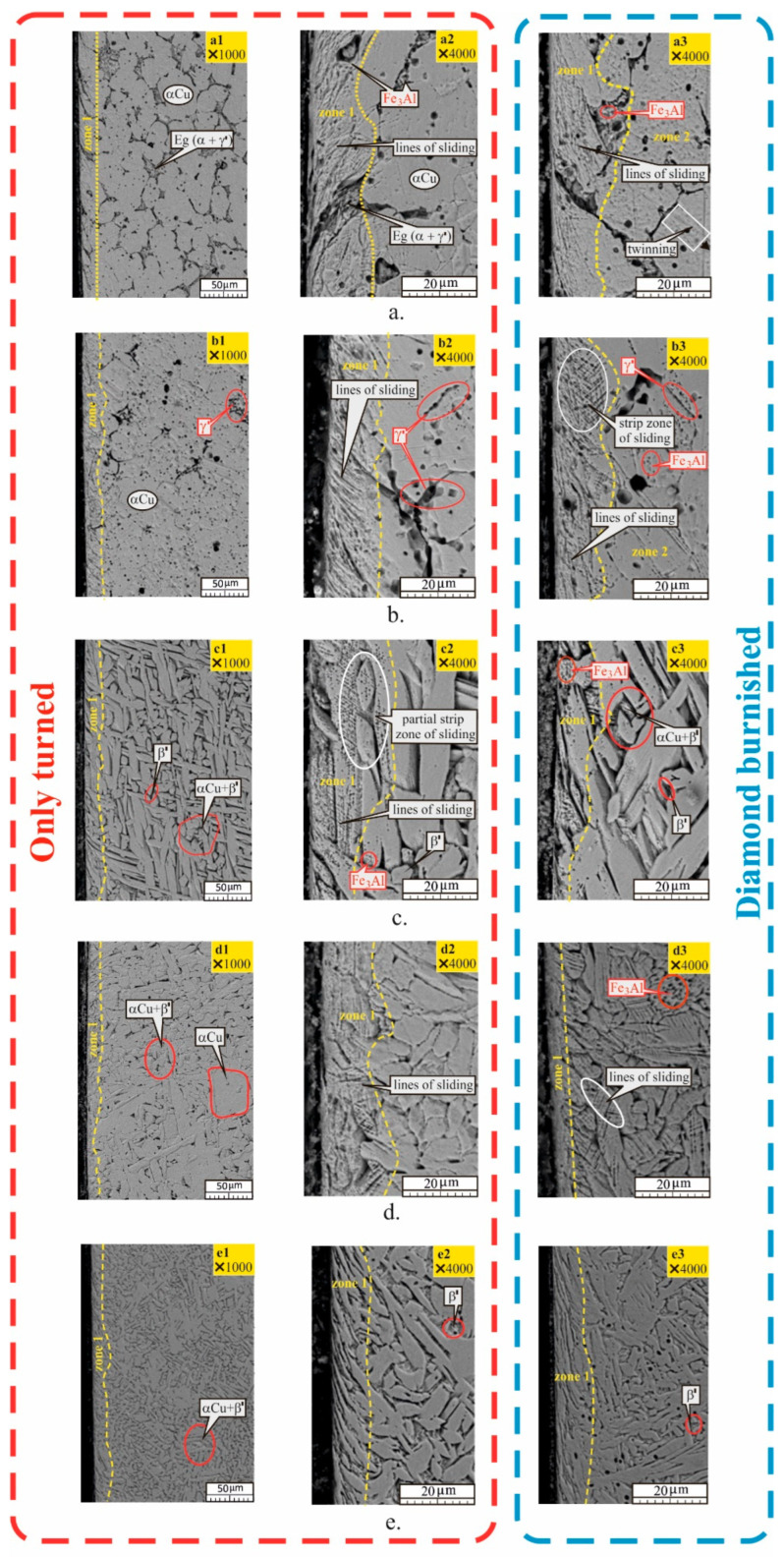
Microstructure of Cu-10Al-5Fe bronze: (**a**) as received; (**b**) annealed; (**c**) quenched; (**d**) quenched and tempered at 600 °C; (**e**) quenched and tempered at 300 °C.

**Figure 7 materials-15-08905-f007:**
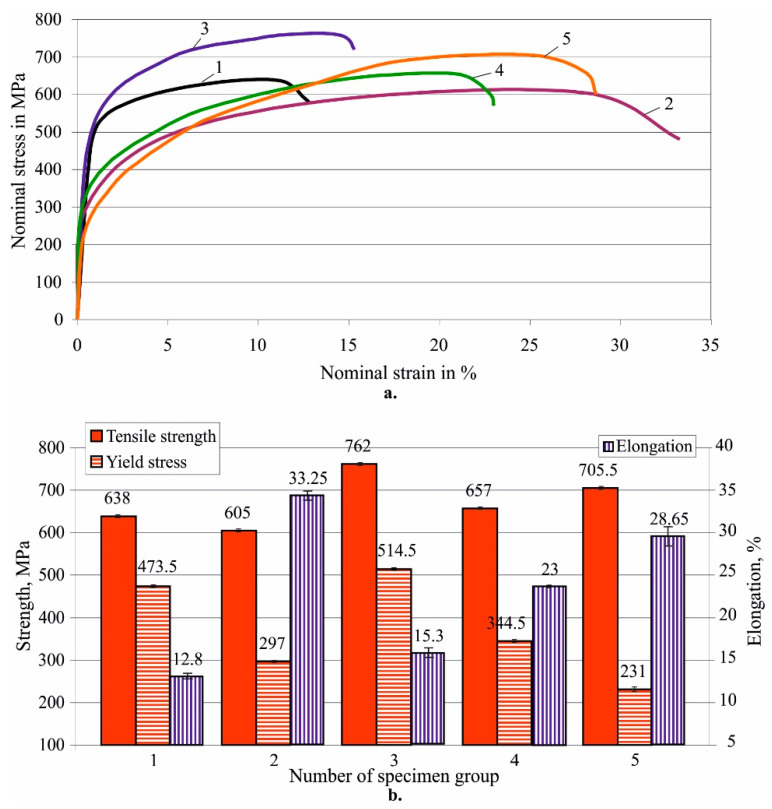
Effect of HT on mechanical characteristics of Cu-10Al-5Fe bronze—tensile strength, yield stress and elongation: (**a**) stress—strain diagrams; (**b**) main mechanical characteristics; 1—as received, 2—annealed, 3—quenched, 4—quenched and tempered at 600 °C, 5—quenched and tempered at 300 °C.

**Figure 8 materials-15-08905-f008:**
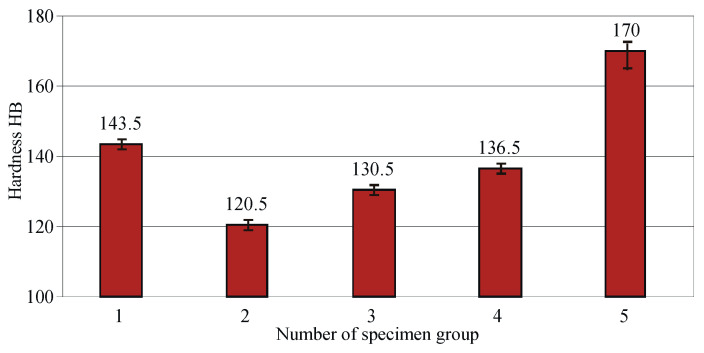
Effect of the heat treatment on hardness of Cu-10Al-5Fe bronze: 1—as received, 2—annealed, 3—quenched, 4—quenched and tempered at 600 °C, 5—quenched and tempered at 300 °C.

**Figure 9 materials-15-08905-f009:**
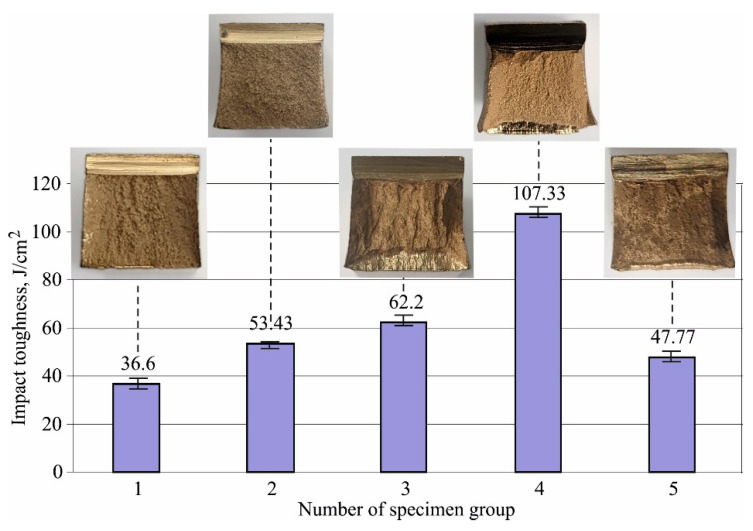
Effect of the heat treatment on impact toughness of Cu-10Al-5Fe bronze: 1—as received, 2—annealed, 3—quenched, 4—quenched and tempered at 600 °C, 5—quenched and tempered at 300 °C.

**Figure 10 materials-15-08905-f010:**
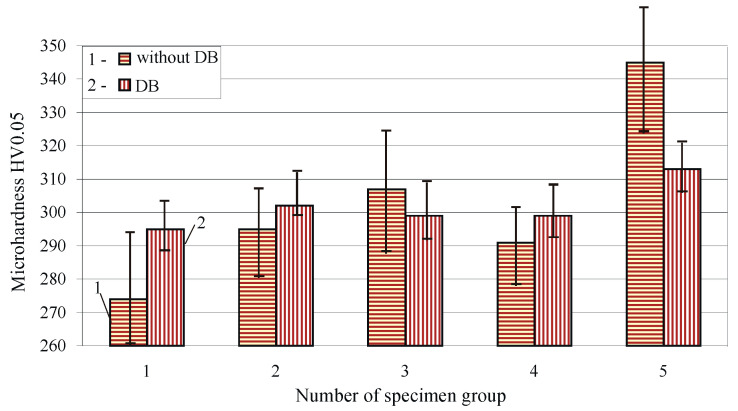
Effect of heat treatment and severe surface plastic deformation on the micro-microhardness of Cu-10Al-5Fe bronze: 1—as received, 2—annealed, 3—quenched, 4—quenched and tempered at 600 °C, 5—quenched and tempered at 300 °C.

**Figure 11 materials-15-08905-f011:**
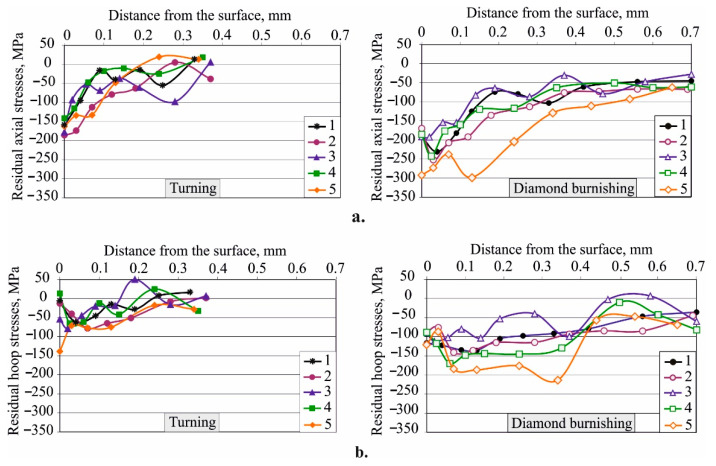
Residual stress distribution: (**a**) axial, (**b**) hoop: 1—as received, 2—annealed, 3—quenched, 4—quenched and tempered at 600 °C, 5—quenched and tempered at 300 °C.

**Figure 12 materials-15-08905-f012:**
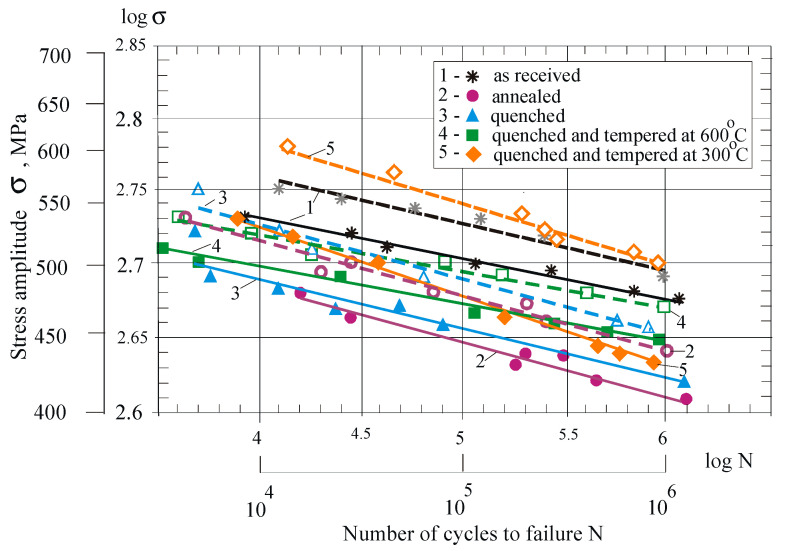
Effect of heat treatment and severe surface plastic deformation on fatigue behaviour of Cu-10Al-5Fe bronze: dashed lines correspond to severe surface plastic deformation via DB.

**Figure 13 materials-15-08905-f013:**
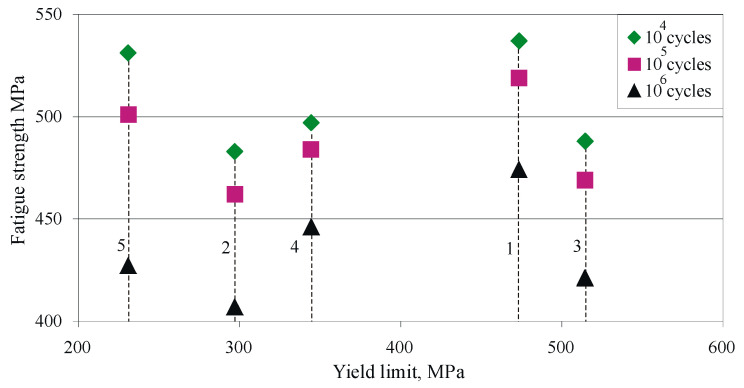
Dependence of fatigue strength on yield limit at different fatigue life: 1—as received, 2—annealed, 3—quenched, 4—quenched and tempered at 600 °C, 5—quenched and tempered at 300 °C.

**Figure 14 materials-15-08905-f014:**
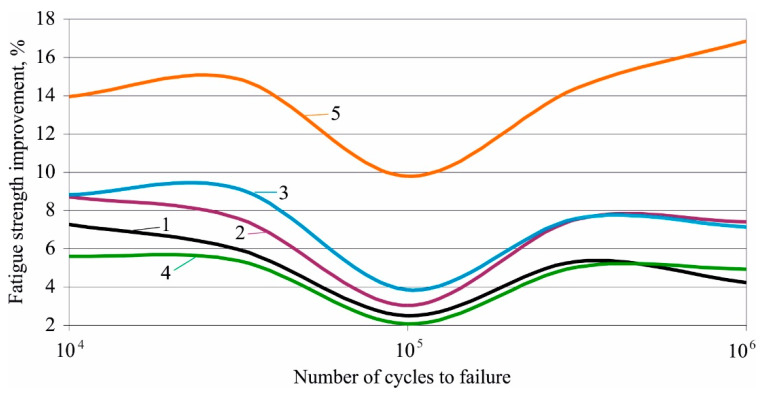
Fatigue strength improvement via DB: 1—as received, 2—annealed, 3—quenched, 4—quenched and tempered at 600 °C, 5—quenched and tempered at 300 °C.

**Figure 15 materials-15-08905-f015:**
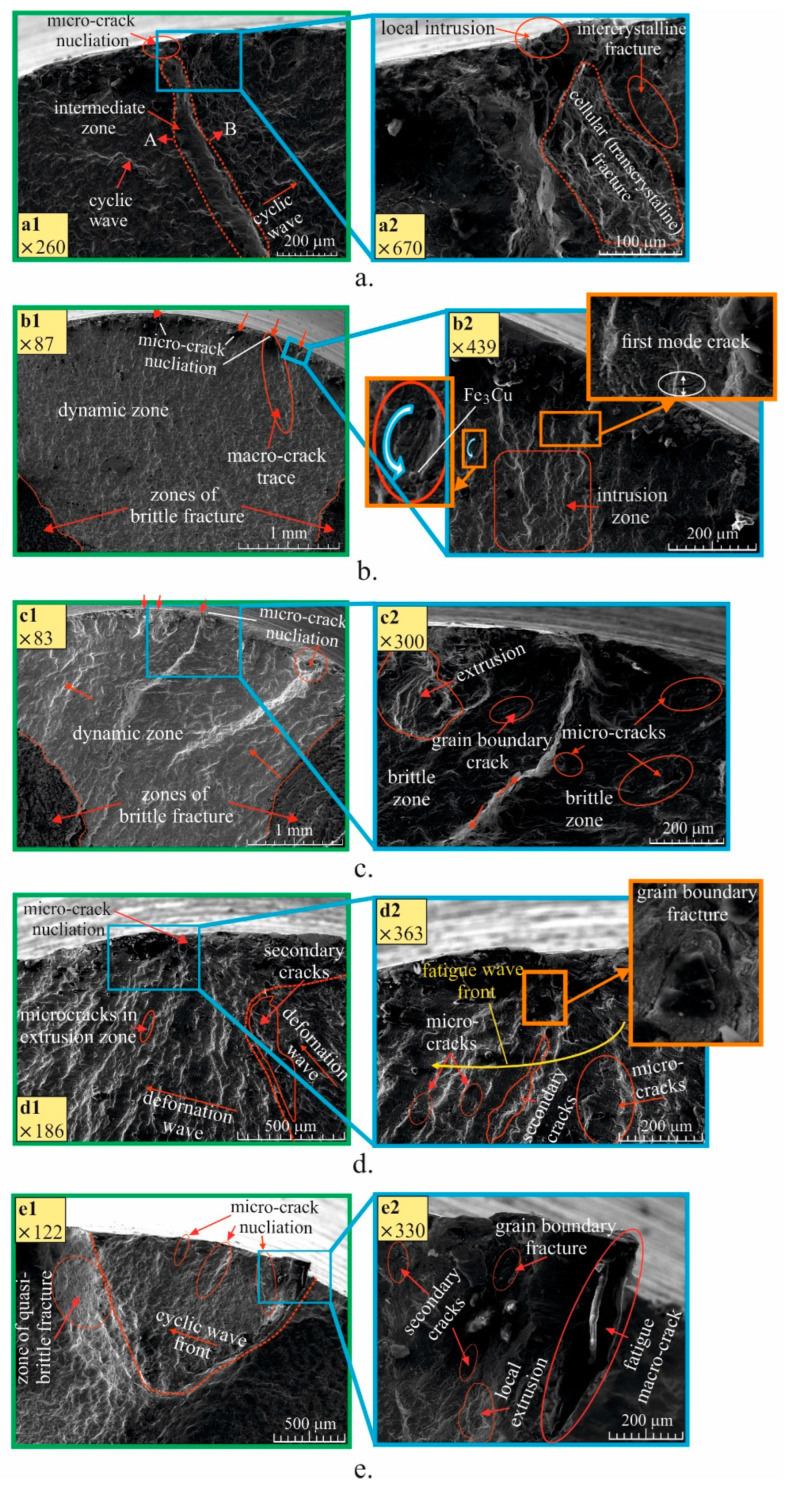
Fracture surfaces of the fatigue specimen heat treated in different manner: (**a**) as received; (**b**) annealed; (**c**) quenched; (**d**) quenched and tempered at 600 °C; (**e**) quenched and tempered at 300 °C.

**Figure 16 materials-15-08905-f016:**
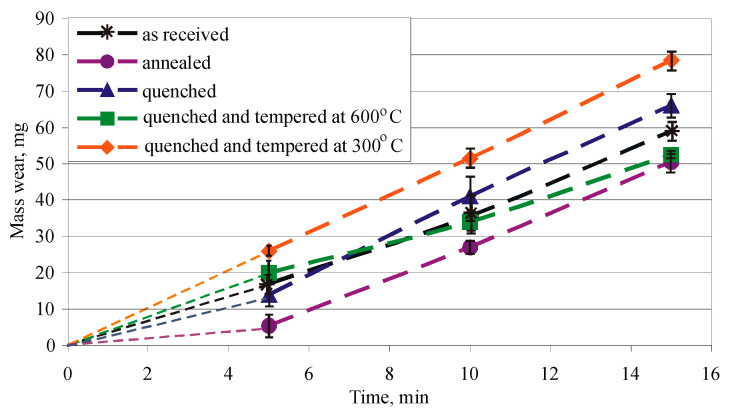
Mass wear: 1—as received, 2—annealed, 3—quenched, 4—quenched and tempered at 600 °C, 5—quenched and tempered at 300 °C.

**Figure 17 materials-15-08905-f017:**
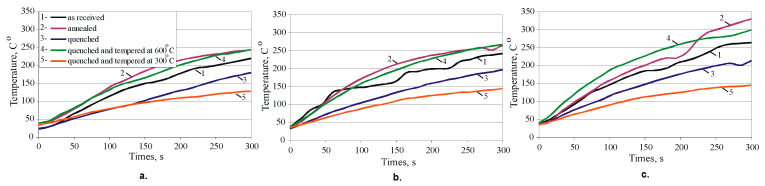
The contact temperature change: (**a**). 0–5min; (**b**). 5–10 min; (**c**). 10–15 min.

**Figure 18 materials-15-08905-f018:**
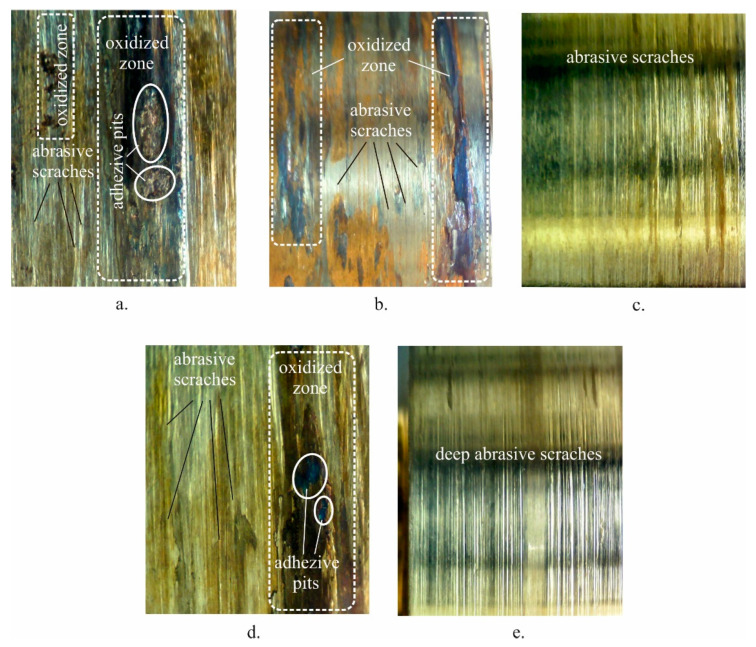
Worn surfaces under dry friction condition: (**a**) as received; (**b**) annealed; (**c**) quenched; (**d**) quenched and tempered at 600 °C; (**e**) quenched and tempered at 300 °C.

**Table 1 materials-15-08905-t001:** Effect of heat treatments on mechanical characteristics.

Ref.	Heat treatment Parameters	Mechanical Characteristics	Bronze
[3]	Optimal HT mode	Hardness	Cu-9Al-4Fe
[5]	Tempering time	Hardness	Cu-9Al-4Fe
[9]	Annealing temperature	Hardness	Cu-Al-Ni-Fe Cu-Al-Fe
[10]	Quenching and tempering at different temperatures	Hardness	Cu-Al-NI-Fe
[11]	Tempering at different temperatures	Hardness; impact toughness	Cu-Al-Fe-Ni Cu-Al-Fe-Mn
[12]	Aging time and temperature	Hardness; tensile strength; elongation	Cu-Al-Fe-Be Cu-Al-Fe-Ni
[13]	Tempering time and temperature	Hardness	Cu-Al-Fe-Ni

**Table 2 materials-15-08905-t002:** Chemical composition in percentages (wt%) of Cu-10Al-5Fe bronze.

Cu	Al	Fe	Mn	Ni	Pb	Zn	Si	Other
81.68	10.6	5.70	0.59	0.42	0.117	0.061	0.022	Balance

**Table 3 materials-15-08905-t003:** Comparison of the effectiveness of the types of heat treatment.

№	Criteria (Requirements)	Ranking of Heat Treatment Types (Sample Groups)				
		I Group	II Group	III Group	IV Group	V Group
1	Static tensile strength	4	5	**1**	3	2
2	Yield limit	2	4	**1**	3	5
3	Plasticity (elongation; cold work)	5	**1**	4	3	2
4	Hardness	2	5	4	3	**1**
5	Surface micro-hardness	5	3	2	4	**1**
6	Impact toughness	5	3	2	**1**	4
7	Fatigue strength	2	5	4	3	**1**
8	Friction	3	5	2	4	**1**
9	Dry sliding wear resistance	3	**1**	4	2	5
Integral efficiency		31	32	24	26	**22**

## Data Availability

Not applicable.

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
