# Peer review of "Effects of Heat Treatment and Severe Surface Plastic Deformation on Mechanical Characteristics, Fatigue, and Wear of Cu-10Al-5Fe Bronze"

_materials, 2022, doi:10.3390/ma15248905_

Round 1
Reviewer 1 Report
1. Abbreviations that appear for the first time in the article (abstract) should be defined.
2. In this article, the “IAB” is not suitable as an abbreviation for iron-aluminum bronzes. Please revise it.
3. In tensile test, the difference between tensile strength and yield stress increased with a decrease in tempering temperature. Why?
4. In Fig. 8, the hardening (heating at 920°C and quenching in water) of the as-received state leads to a lower hardness than the original. Why?
5. Authors indicated that low-temperature ageing decreases the hardness less, which allows for brittleness to remain in the material (Fig. 9). This make me confuse.
6. What is the contribution of residual compressive stress to the Cu-10Al-5Fe alloy? It does not seem to be mentioned in the article.
7. Is the fatigue limit of the Cu-10Al-5Fe with different heat treatments obtained in the S-N curves?
8. Throughout conclusion needs to be revised.
9. The manuscript requires a better improvement in terms of language and grammar usage.

Reviewer 2 Report
1. Better to mention the details of the standards of the tests performed for this work.
2. It was mentioned, "Several samples reached the fatigue strength limit (10^7cycles) without failing, after which the test was terminated". Then how the authors could make concluding marks on fatigue?
3. It is essential to provide counter-body details such as properties like hardness, as a part of the fretting wear test, as it is very influential.
4. Rather than or apart from giving a photograph in Fig.3, it is advisable to provide a schematic line diagram of the setup.
5. Mechanical properties are provided in Fig.7, Further it is strongly suggested to provide few representative stress-strain diagrams.
6. From Figs. 7 and 8, it is shown that quenched and tempered condition do exhibits higher hardness than that of as quenched condition. The same is not true for tensile strength. Further no correlation between yield strength and tensile strength between these two conditions. Need a details explanation.
7. In contrast to the above point, as per Fig.10, the hardness values are showing a reverse trend after severe surface plastic deformation. Why?
8. In the experimental procedure, it is mentioned that tests are conducted till 10^7 cycles, whereas the data provided in Fig.12 is not up to that. Why?
9. abnormally high number of figures. It is suggested to reduce the total number of figures (if possible) by combining some sets of testing data.
10. It is advised to avoid the initials of the authors while citing in the running manuscript such as in line number 66.
Reviewer 3 Report
This manuscript investigated the effects of different heat treatments and surface plastic deformation of Cu-10Al-5Fe IAB on the microstructure, surface integrity, mechanical properties, fatigue strength, and wear resistance. To improve the quality of the manuscript, some comments are suggested as follows:
1. The language can be improved by correct the spelling and sentence structure. In addition, more accurate descriptions and quantitative discussions should be made especially for the abstract and conclusion.
2. DB should be defined at the first sight in the Abstract.
3. There are too many paragraphs in the introduction. The authors are suggested to re-organize the structure by merging some of the related paragraphs.
4. Why did the authors choose the five group of heat treatment with different durations and cooling methods? One refence is suggested by considering the homologous temperature: Cooling and annealing effect on indentation response of lead-free solder.
5. The details should be provided to measure the hardness in Section 2.4.3. What is the type of the indenter? What is the penetration depth to determine the hardness?
6. Is there any relationship between the ratio of yield strength and ultimate tensile strength after different HTs?
7. Please further discuss the effect of HT on the yield strength and hardness, as there is an intrinsic relationship between yield strength and hardness. But it is abnormal for the cases 4 and 5.
8. Some optimal HT are suggested. It is read that ‘For example, if the component is designed for rotational bending conditions, the combination of quenching at 920°C in water, subsequent tempering at 300°C for three hours’. Is it possible to propose some combinations of HT different from the five cases? This could be promising for wider industrial applications.
9. The quality of Fig. 11 should be improved for better clarity.
10. The highlight in green in Table 3 is inappropriate. This can be modified to be bold or italic.
11. The conclusions are messy, which can be condensed as two paragraphs.
12. Page numbers are missing for some of the references.
Round 2
Reviewer 2 Report
The authors have done all necessary suggested modifications.
Author Response
Reviewer 2
We would like to thank the Reviewer 2 for his comment.
Comment
“The authors have done all necessary suggested modifications.”
Answer
Thank you very much for your comment and your rating for our article.
Reviewer 3 Report
Accept after minor revision of language
Author Response
Reviewer 3
We would like to thank the Reviewer 3 for his comment.
Comment
“Accept after minor revision of language”
Answer
Тhe article was subjected to professional proofreading (British English) by Cambridge Proofreading LLC. Please, see the attached file.
